# Pharmacokinetics of Antibiotics in Pediatric Intensive Care: Fostering Variability to Attain Precision Medicine

**DOI:** 10.3390/antibiotics10101182

**Published:** 2021-09-28

**Authors:** Matthias Gijsen, Dirk Vlasselaers, Isabel Spriet, Karel Allegaert

**Affiliations:** 1Clinical Pharmacology and Pharmacotherapy, Department of Pharmaceutical and Pharmacological Sciences, KU Leuven, 3000 Leuven, Belgium; isabel.spriet@uzleuven.be (I.S.); karel.allegaert@uzleuven.be (K.A.); 2Pharmacy Department, UZ Leuven, 3000 Leuven, Belgium; 3Laboratory of Intensive Care Medicine, Department of Cellular and Molecular Medicine, KU Leuven, 3000 Leuven, Belgium; dirk.vlasselaers@uzleuven.be; 4Clinical Division of Intensive Care Medicine, UZ Leuven, 3000 Leuven, Belgium; 5Department of Development and Regeneration, KU Leuven, 3000 Leuven, Belgium; 6Department of Hospital Pharmacy, Erasmus MC University Medical Center, 3015 GD Rotterdam, The Netherlands

**Keywords:** pediatric, antibiotic, critical illness, pharmacokinetics, augmented renal clearance, extracorporeal membrane oxygenation, continuous renal replacement therapy, cardiopulmonary bypass, whole body hypothermia, therapeutic drug monitoring

## Abstract

Children show important developmental and maturational changes, which may contribute greatly to pharmacokinetic (PK) variability observed in pediatric patients. These PK alterations are further enhanced by disease-related, non-maturational factors. Specific to the intensive care setting, such factors include critical illness, inflammatory status, augmented renal clearance (ARC), as well as therapeutic interventions (e.g., extracorporeal organ support systems or whole-body hypothermia [WBH]). This narrative review illustrates the relevance of both maturational and non-maturational changes in absorption, distribution, metabolism, and excretion (ADME) applied to antibiotics. It hereby provides a focused assessment of the available literature on the impact of critical illness—in general, and in specific subpopulations (ARC, extracorporeal organ support systems, WBH)—on PK and potential underexposure in children and neonates. Overall, literature discussing antibiotic PK alterations in pediatric intensive care is scarce. Most studies describe antibiotics commonly monitored in clinical practice such as vancomycin and aminoglycosides. Because of the large PK variability, therapeutic drug monitoring, further extended to other antibiotics, and integration of model-informed precision dosing in clinical practice are suggested to optimise antibiotic dose and exposure in each newborn, infant, or child during intensive care.

## 1. Introduction

Antibiotics are commonly used in hospitalised children [1,2,3]. A one-day point-prevalence study reported antibiotic use in approximately 40% of all hospitalised children in European and non-European hospitals [1]. Another retrospective cohort study of 40 US children’s hospitals found that 60% of all children received at least one antibiotic during their hospital stay, representing up to 601 antibiotic days per 1000 patient days [2]. The highest rates of antibiotics use have been reported in critically ill children and patients admitted to the hemato-oncology ward [1]. Up to 57–79% of all patients received antibiotics during their stay at the pediatric intensive care unit (PICU) [4]. Of the five drugs most commonly administered to neonates admitted in a neonatal ICU (NICU), three are antibiotics (ampicillin, gentamicin, and vancomycin) [5].

The decision on the timing of prescription (when), the choice of the antibiotic(s) (what), and appropriate dosing (how much) to achieve sufficient exposure are crucial determinants for successful antibiotic therapy [6]. Pharmacokinetics (PK) and its variability mainly relates to the ‘how much’ question. It is commonly accepted that adequate exposure above a critical target or within a given range is required for successful (effective and safe) antibiotic therapy [7]. Hence, knowledge of the processes that lead to drug exposure after its administration is essential to select the most appropriate dosing regimen. These processes consist of absorption, distribution, metabolism, and excretion to characterise the pharmacokinetics of a drug. In the setting of pediatric or neonatal intensive care and critical illness, these processes are modulated by both maturational (age, weight) as well as non-maturational (disease, therapeutic interventions) covariates.

Children show important developmental or maturational changes, which may contribute greatly to PK variability observed in pediatric patients [8,9]. Standard weight-based dosing regimens have been shown to lead to suboptimal exposure in certain pediatric patient populations [10,11], so that organ function should be taken into account to optimise pediatric drug dosing, also when the focus is on antibiotics [12,13]. These PK alterations are further enhanced by disease-related or non-maturational factors. Specific to the intensive care setting, such factors include critical illness, inflammatory status, augmented renal clearance (ARC), as well as therapeutic interventions (e.g., extracorporeal organ support systems or whole-body hypothermia [WBH]) (Figure 1) [14].

Most drugs—also antibiotics—have been developed in and for adults, with subsequent extrapolation of dosing regimens to children. Pediatric antibiotic trials are seldom performed due to ethical, practical, and economic considerations [8]. This is evidenced by a recent report of Thaden et al., who reviewed all pediatric trials registered in clinicaltrials.gov between October 2007 and September 2017 [15]. Less than 1% of all pediatric trials (122/17,495) were industry or US government-funded trials investigating systemic antibacterial or antifungal drugs. Consequently, off-label (dose, indication, formulation) prescription is very common, be it that recommendations on how health care professionals can handle this setting have recently been published [16,17]. It has been suggested that certain pediatric patient populations may benefit from individualised dosing to increase target attainment, while minimizing the risk of unnecessary toxicity, using model-informed precision dosing (MIPD) concepts in these populations [11,18].

The aim of this narrative review is to illustrate the relevance of maturational changes in absorption, distribution, metabolism, and excretion (ADME) applied to antibiotics, and to perform a focused assessment of the available literature on the impact of critical illness—in general, and in specific subpopulations (ARC, extracorporeal organ support systems, WBH)—on PK and potential underexposure in children and neonates. Such information on the relevant PK covariates is crucial to implement precision medicine for antibiotics in NICU and PICU populations. We, therefore, conducted a PubMed search to identify all relevant English-language articles and abstracts reporting antibiotic PK data in critically ill children and neonates from inception until July 2021. Two authors performed the literature review (M.G. and K.A.).

## 2. Developmental Pharmacokinetics, Applied to Antibiotics

Developmental pharmacology describes the impact of maturation on drug disposition (PK) and drug effects (pharmacodynamics [PD]) throughout the pediatric age range (0–18 y) [19]. There is regulatory guidance to support pediatric drug development within a generic framework, also applicable to antibiotics [20]. This framework is based on 3 pillars, related to: (i) disease progression similarity between adults and children; (ii) the similarity in response to intervention between both, and (iii) the availability of relevant and valid PD markers (biomarkers, outcome variables). Applying this framework to antibiotics, regulatory bodies currently consider it to be reasonable to assume similarity on antimicrobial PD between patient populations (concentration-response) because the treatment is aimed at the infectious organism, not the host (adult or child) [19,20]. Consequently, differences in PK and safety are the primary focus to optimise drug development programs and antibiotic utilization in neonates and children.

A recent ‘throughout-life’ PK modelling effort on vancomycin hereby illustrated that the current dosing recommendations (Food and Drug Administration label) commonly resulted in underexposure, with the highest probability in children and adolescents (70%), except in newborns, while the label remains very descriptive on dosing in preterm neonates [21,22].

Absorption: the impact of gastrointestinal variability on oral drug absorption and the relevance of developmental changes (gastric pH, gastric emptying, gastric fluid composition, intestinal and colonic transit time, first-pass metabolism, pancreatic function) has recently been reviewed [23]. A systematic review recently reflected on these changes and assessed the available evidence on the switch from parenteral to oral antibiotics in neonates. Based on 31 studies and compared to parenteral administration, oral antibiotics generally reached maximum concentration later and had a lower bioavailability. However, in the majority of studies, adequate serum levels were reached, so that early switch could be beneficial when considering health economics and the burden to the patient and family [24].

Distribution: throughout infancy, the extracellular and total body water proportions are higher, and this results in higher volumes of distribution (Vd) and lower (peak) concentrations for water-soluble antibiotics (e.g., aminoglycosides, vancomycin, beta-lactams) when administered on an mg/kg basis. Table 1 summarises maturational changes in body water composition, while Figure 2A illustrates the impact of the body water composition on the Vd (L/kg) for a water-soluble antibiotic (amikacin) [25,26,27,28]. These patterns can be further affected by disease-related aspects, similar to those observed in adults, as, e.g., Lingvall et al. described that the Vd of gentamicin was significantly higher (+14%) in septic compared to non-septic neonates [29]. Table 1 also summarises the maturational trends in plasma protein composition, as this determines the drug-binding capacity and free concentrations, as illustrated for, e.g., cefazolin or vancomycin [30,31,32]. Competitive binding of co-administered drugs or endogenous substances (indirect bilirubin) may occur and explains the contraindication for ceftriaxone in neonates [33].

Elimination: the main route of elimination of most antibiotics is by renal excretion, determined by both glomerular filtration rate (GFR) and renal tubular activities (secretion, reabsorption) so that pharmacometric approaches to personalise the use of primarily renally eliminated antibiotics in a pediatric and neonatal ICU setting should focus on renal models and renal (patho)physiology [13]. A maturational human renal function with a sigmoid hyperbolic model has been described so that half of the adult value is reached at 48 weeks postmenstrual age to be at 90% of this value at the end of infancy, even when an allometric coefficient (kg^0.75^) was used [34]. When based on body weight, the steep increase in clearance (CL) in infancy is even more pronounced. Figure 2B illustrates this maturational pattern in CL (L/kg/h), based on a published amikacin dataset [28]. Along the same line, and based on datasets of gentamicin, tobramycin, and/or vancomycin, semi-physiological functions for GFR maturation throughout human life have been constructed [21,35]. When scaling of CL was based on these GFR maturational models, maturational changes in plasma protein concentration only marginally affected renal clearance, except for drugs highly bound to alpha-1-acid glycoprotein [36]. In contrast, the contribution of active tubular secretion to overall renal clearance can range—depending on drug-specific PK properties—between 41 and 90%, so that ontogeny of tubular secretion cannot be ignored, not even in infants [37].

Such semi-physiological functions are relevant to truly assess the impact of non-maturational factors. Besides aspects related to the critical illness that will be discussed below, this also includes co-administration of nephrotoxic drugs, like non-steroidal anti-inflammatory drugs (ibuprofen, indomethacin), commonly administered to induce closure of a patent ductus arteriosus. Using the impact of those drugs on aminoglycoside or vancomycin CL, the transient reduction in GFR during ibuprofen or indomethacin administration is −20 and −40%, respectively, in the newborn [38,39].

## 3. Critical Illness Related Pharmacokinetic Alterations in Children and Infants

Pathophysiological changes and resulting PK alterations have been demonstrated for many antibiotics in critically ill children. Nevertheless, pediatric PK data remain relatively scarce and are lagging behind the plethora of PK studies reporting antibiotic PK in critically ill adults. Increased Vd is frequently observed due to fluid resuscitation and inflammation-induced capillary leak, further enhanced by hypoalbuminemia. Alterations in antibiotic CL have also been reported repeatedly in pediatric antibiotic studies, and have been related to both extremes of renal function (i.e., acute kidney injury necessitating renal replacement therapy [RRT] and ARC) [40,41]. Despite the similarity to patterns in critically ill adults, still, important differences in PK alterations have been reported in pediatric patients, as their phenotypic PK findings are the result of merged maturational (age- or weight-related) and non-maturational factors, like critical illness.

Higher Vd (L/kg) have been reported for most beta-lactam antibiotics in pediatric patients with severe bacterial infections compared with adult ICU patients. However, it should be noted that there is still large variability in PK between and within the different pediatric studies [40,41]. Bodyweight and age-related covariates are the most frequently retained predictors of interindividual PK variability on Vd and CL in pediatric ICU studies. Whereas renal function is the most frequently identified covariate for PK alterations in adult ICU studies, it is less frequently identified in pediatric ICU studies. This suggests that markers for renal function (which are mostly based on serum creatinine [SCr]) are less performant in predicting changes in the CL of antibiotics in pediatric vs. adult ICU patients. This is probably at least partly due to age-related changes in renal function. This maturational variability to a certain extent blurs non-maturational effects, which is a phenomenon that is most pronounced in early neonatal life and infancy [13,14]. Although renal function (GFR and tubular secretion) is lower in neonates and young infants, weight-corrected (per kg) GFR values are much higher (up to 70%) in young children (2–5 years), compared to adults [42].

Despite >50 published antibiotic PK studies in critically ill pediatric patients, PK data remain scarce and for most antibiotics, PK knowledge is based on only a few studies with small sample sizes [40,41]. For some commonly used antibiotics, there are even no PK data available in critically ill children (e.g., ceftazidime). Overall, PK studies show that the attainment of predefined PK/PD targets is suboptimal with standard dosing of many antibiotics in critically ill children [40,41,43]. Therefore, several studies suggest dose optimization strategies for these patients. Interestingly, the same PK/PD targets are commonly aimed for in pediatric and adult critically ill patients [11,41,44]. Table 2 presents an overview of the three categories of PK/PD targets commonly used and the antibiotic classes falling into each category [11,44,45]. Next to the classic advice to increase the dose or dosing frequency, extended or continuous infusion strategies are regularly suggested to increase the probability of target attainment (PTA) with beta-lactam therapy in critically ill children. This strategy has been extrapolated from adult critically ill patients, however, until now, there is no conclusive evidence of its clinical benefit. Interestingly, a subgroup analysis of critically ill children included in a recent retrospective chart analysis revealed a potential mortality benefit (2.1% vs. 19.6%) of extended vs. intermittent beta-lactam infusion [43]. Additionally, therapeutic drug monitoring (TDM) is regularly suggested to overcome suboptimal exposure in critically ill children, also for beta-lactams. TDM is an interesting strategy for dose optimization considering the high inter-individual PK variability observed in these patients. Cies et al. found a relatively low mortality rate of 4.3–12.2% in pediatric ICU patients undergoing routine beta-lactam TDM and dose adjustments according to predefined PK/PD targets [46]. Compared to mortality rates of 25% or higher observed in septic pediatric patients [47,48,49], this study suggests a mortality benefit from beta-lactam TDM in these patients. Notwithstanding, confounding from baseline differences in patient characteristics cannot be excluded.

It should be noted that most PK studies report antibiotics that are subjected to routine TDM in clinical practice (e.g., vancomycin and aminoglycosides). Table 3 summarises all antibiotic PK studies performed in critically ill children. Studies performed only in neonates (or adults) were excluded from the general summary presented in Table 3, as these fell out of the scope of the current summary.

Vancomycin is by far the most frequently reported antibiotic in critically ill children, with studies covering its PK over the whole pediatric age range in more than 1000 critically ill children [50,51,52,53,54,55,56,57,58,59,60,61,62,63,64,65,66,67,68]. Teicoplanin, another antibiotic belonging to the same antibiotic class as vancomycin, has been described in four studies [69,70,71,72]. Three of these four studies reported insufficient target attainment with standard dosing regimens [70,71,72]. Furthermore, the most recent study reported highly variable unbound teicoplanin concentrations, highlighting the need for TDM of unbound teicoplanin [72].

Aminoglycosides are also commonly monitored in clinical practice, which is reflected by several studies describing the PK of gentamicin [73,74,75,76] and amikacin [77,78,79]. For gentamicin, higher initial doses and longer dosing intervals have been suggested for neonates and (young) infants in two studies to compensate for an (age-related) increase in Vd and decrease in CL, respectively [74,75]. Similar findings have been reported for amikacin [77,79].

Beta-lactam PK studies in critically ill children show diverse results and are mostly limited to a few studies or cases reports for a specific beta-lactam antibiotic [46,81,82,83,84,85,86,87,88,89,90,91,92,93,94,95,96,97,98,99,100,101]. In general, studies found suboptimal exposure with standard beta-lactam dosing regimens, due to an increased beta-lactam CL rate. Several studies recommended increased dosing and/or prolonged infusion to achieve target attainment [81,82,83,84,87,91,93,95,97,98,99,101].

Finally, we also found a few PK studies describing linezolid [102], ciprofloxacin [103], and daptomycin [104,105,106] PK in critically ill children (Table 3).

## 4. Subpopulations of Critically Ill Children and Neonates at Increased Risk of Pharmacokinetic Alterations

### 4.1. Augmented Renal Clearance

Although body weight and age-related covariates are most consistently identified in antibiotic PK studies in children, renal function is also frequently retained as a significant predictor of antibiotic exposure and target attainment. While decreased renal function is generally taken into account in drug dosing guidelines, increased renal function was rarely considered until recently. ARC, referring to the enhanced renal elimination of solutes (e.g., drugs), has only been identified as a major predictor of altered drug CL over the past decade [107]. The vast majority of studies were performed in adult ICU patients. However, a recent study confirmed that ARC is also highly prevalent in critically ill children aged 1 month to 15 years old (67% of patients showed ARC at least once during a 4-day study period) [108]. Importantly, it has been recommended to use age-related threshold values to detect ARC in children, as reference values for GFR evolve at the beginning of life to reach a plateau at 2 years of age [108]. These threshold values range from 70 mL/min/1.73 m^2^ at 1 month of postnatal age to 150 mL/min/1.73 m^2^ for all children aged 2 years and older [109].

Similar to previous findings in adults, there is an increasing number of studies demonstrating subtherapeutic exposure to antibiotics with predominant renal clearance in pediatric patients with ARC. The majority of these studies have been performed in critically ill children, and consistently reveal increased antibiotic CL associated with increased renal function, mostly measured by means of the estimated GFR_Schwartz_ or the SCr concentration [110]. An overview of all studies investigating antibiotic PK in critically ill children and neonates with ARC is provided in Table 4.

Most antibiotic PK studies in ARC patients have been performed with beta-lactams [81,82,83,84,87,91,93,95,97,98]. Considering their time-dependent antimicrobial activity, beta-lactams are probably most susceptible to suboptimal PK/PD target attainment due to ARC. As mentioned above, dose optimisation strategies suggested by most authors consisted of extended or continuous infusion to achieve target attainment.

Several studies have also reported subtherapeutic vancomycin exposure in critically ill children with ARC [52,56,64,111,112,113,114,115]. Increased loading or initial doses have been recommended to achieve early therapeutic exposure. Two studies identified patients with febrile neutropenia and/or hemato-oncologic diseases as being at increased risk of ARC and subtherapeutic exposure [56,112]. A recent study based on a pooled population PK and with a focus on dosing optimisation of vancomycin in young infants, children, and adolescents (0.4–14.9 years) further illustrated and quantified the relevance of ARC [115]. The authors hereby concluded that in the pediatric ARC setting, the current recommended vancomycin dose of 60 mg/kg/day was associated with a high risk of underdosing. To reach the target AUC/MIC of 400–700 in these pediatric patients, the vancomycin dose should be increased to 75 mg/kg/day (+25%) for infants and children between 1 month and 12 years of age, and to 70 mg/kg/day (+18%) for adolescents between 12 and 18 years of age.

Interestingly, even for concentration-dependent antibiotics like gentamicin, ARC has been shown to contribute to subtherapeutic exposure. A recent study demonstrated decreased exposure (AUC_0–24_) and target attainment for gentamicin in critically ill children with ARC vs. those without ARC [76]. As the bactericidal activity has been reported to correlate well with the AUC of gentamicin in preterm neonates, the authors recommend measuring the AUC of gentamicin in critically ill children with ARC. Another study found that increased amikacin doses are needed in burns patients with sepsis due to increased CL and Vd [116].

Even though many antibiotics show predominant renal elimination, renal function was only identified as a significant predictor in a limited proportion of the existing PK studies in pediatric patients. Most PK studies have been performed in critically ill children. It might be that the currently applied covariates to monitor renal function (SCr as such, or SCr-based estimated GFR or creatinine clearance) are suboptimal markers of renal function in this population [41,108]. SCr is dependent on several demographic and disease-related factors, which lead to rapidly changing concentrations, especially in critically ill patients who are, per definition unstable. Moreover, SCr reflects the creatinine production and excretion over the past 24 h [41]. Consequently, its value will always be lagging behind in patients with rapidly changing renal function. Therefore, special emphasis has been laid on the need for a non-invasive covariate that allows reliable and continuous monitoring of the fluctuating renal function in critically ill children [110]. Until now, we are not aware of any paper on ARC in (pre)term neonates, perhaps because this subpopulation already displays a very steep maturational increase in GFR and renal clearance (reflected in Figure 2B), so that it is easier to observe a delay in this pattern, like, e.g., in the event of ibuprofen or indomethacin co-exposure or during WBH [38,39].

### 4.2. Extracorporeal Organ Support Systems

#### 4.2.1. Extracorporeal Membrane Oxygenation

Extracorporeal membrane oxygenation (ECMO) is a life-saving technique used as a bridge-to-recovery for pediatric and adult patients with severe respiratory and/or cardiac failure [117]. ECMO has been identified both as a cause and consequence of bacterial infections [118]. The infection rate is 9–14% in pediatric patients supported with ECMO. Unsurprisingly, antibiotic use is high in ECMO patients [117]. It has been suggested that ECMO may lead to additional PK alterations during antibiotic therapy in critically ill patients [117,118,119].

The potential impact of ECMO on antibiotic PK can be related to the direct interaction of the antibiotic with the ECMO circuit (e.g., adsorption to the tubing or oxygenator) [117,118,119]. Ex vivo experiments revealed that drugs with a higher degree of lipophilicity and drugs with high plasma protein binding are most at risk of sequestration within the ECMO circuit [118,120]. Additionally, when used in patients, ECMO can also increase the Vd of the antibiotic with or without altering its CL. Clinical studies reporting antibiotic PK during ECMO have been performed mostly in neonates, although data in older children and adults are increasingly being reported over the last decade [117,118,119]. Nevertheless, data are scarce, and it remains difficult to translate the scarce data to recommendations for clinical practice.

An increase in Vd is observed for most hydrophilic antibiotics during ECMO in children. The impact of ECMO on Vd is inversely related to age. Due to the smaller amount of circulating blood volume, ECMO-induced hemodilution will have a larger impact in younger children than in older children or adults. Additionally, neonates and infants have a proportionally higher body water composition, which further enhances the impact of ECMO on the Vd of hydrophilic antibiotics [117,119].

As presented in Table 5, most clinical data during ECMO has been collected for vancomycin [121,122,123,124,125,126] and gentamicin [127,128,129,130,131,132], as these are often subjected to routine TDM in clinical practice.

Recent neonatal studies showed conflicting data concerning the impact of ECMO on vancomycin CL [121,122]. Nevertheless, variability in CL was consistently associated with renal function. Hence, current dosing based on age and renal function seems reasonable for vancomycin therapy in pediatric patients supported with ECMO, with close follow-up of plasma concentrations by TDM. For teicoplanin, there are no clinical data in pediatric patients, but extrapolation from adult data suggests a need for higher dosing during neonatal ECMO due to hemodilution and higher plasma protein binding [133].

For gentamicin, a hydrophilic drug with a low Vd, a few studies reported an increased Vd in neonates due to ECMO, which is a logical consequence of more pronounced hemodilution in younger children [128,131]. On the other hand, decreased CL was also observed in several studies, which seemed to be related to decreased renal function in ECMO patients [127,128,130,131,132]. Hence close follow-up of kidney function and TDM should be considered in these patients to guide further dosing.

Increased Vd was also observed during ECMO for cefotaxime [134], cefepime [135], and piperacillin [136] in neonates and infants, while CL was unchanged, decreased, or could not be compared for cefotaxime, cefepime, and piperacillin, respectively. For meropenem, several studies did not demonstrate any significant changes in CL due to ECMO [97,98,137,138]. Two case reports reported altered meropenem CL during ECMO, however, this was probably related to concomitant RRT in both patients [139,140]. For ticarcillin, decreased CL was reported in two adolescents supported with ECMO [141].

Finally, one popPK study in 63 critically ill children did not find a significant difference in linezolid PK in two patients supported with ECMO as compared to the other (non-ECMO) patients [102].

Overall, the effect of ECMO on antibiotic PK in children appears to be limited. Changes in CL can mostly be related to renal function. Furthermore, increased Vd is mainly observed in neonates due to a proportionally higher hemodilution effect of the priming volume associated with ECMO.

#### 4.2.2. Cardiopulmonary Bypass

A few studies describe the PK of the most commonly used antibiotics during cardiopulmonary bypass (CPB). CPB is a technique very similar to ECMO, though only applied for a short period of time, typically the duration of cardiac surgery. Hence, PK alterations similar to those encountered with ECMO may be relevant. Table 6 presents an overview of all antibiotic PK studies performed in children and neonates supported with CPB.

A significant effect from CPB on cefazolin [142] and cefuroxime [143] PK has been reported in popPK studies. These studies have suggested optimised dosing regimens to achieve sufficient PTA in pediatric patients undergoing cardiac surgery with CPB. Another study showed increased Vd and decreased CL of gentamicin during and after CPB use with cardiac surgery [144].

#### 4.2.3. Continuous Renal Replacement Therapy

Approximately 25% of the children admitted to the ICU will develop acute kidney injury, and 4–6% will need RRT [145,146]. Currently, most ICUs apply RRT as continuous veno-venous hemofiltration with (CVVHDF) or without (CVVH) dialysis [146]. It has been suggested that continuous RRT (CRRT) may also contribute to additional antibiotic PK variability in critically ill patients. Currently, a priori dose reductions based on drug dosing guidelines or labels are often performed in CRRT patients [147,148]. However, it has been reported that this strategy leads to inappropriate exposure in many of these patients. Therefore, a more ‘patient-centered approach’ should be considered [148]. A recent observational multinational study showed highly variable antibiotic concentrations in adult critically ill patients receiving RRT. A substantial proportion of patients (>25%) did not achieve optimal exposure according to predefined PK/PD targets for meropenem, piperacillin/tazobactam, and vancomycin. In accordance with other smaller studies in adults and children, this study recommended TDM in critically ill patients receiving CRRT due to the large PK variability measured in these patients [147]. Interestingly, beta-lactam TDM in adult CRRT patients has been shown to lead to improved target attainment [149].

It should be noted that the vast majority of PK studies investigating CRRT were performed in adults. Accordingly, none of the previously published reviews describing PK alterations and antibiotic dosing in CRRT patients provided recommendations for antibiotic therapy in children or neonates receiving CRRT. As a result, recommendations concerning antibiotic dosing in children during CRRT are largely extrapolated from adult studies. It is generally assumed that hydrophilic antibiotics with small molecular weight, and low plasma protein binding are most susceptible to CRRT induced PK alterations [138,148].

The effect of CRRT on antibiotic PK has only been investigated in a few pediatric studies with relatively small sample sizes, as shown in the overview of antibiotic PK studies during CRRT in Table 7. Only one study included more than 15 children receiving CRRT [150]. This recent popPK study demonstrated that vancomycin dosing based on fat-free mass was most appropriate in children supported with CVVHDF. Nevertheless, the most optimal dosing regimen of 15 mg/kg/dose still led to target attainment in only approximately 80% of all patients, due to the high variability in vancomycin PK during CRRT. The second study reporting vancomycin PK during CRRT, a case series, showed that the addition of vancomycin to the CRRT solution resulted in adequate plateau concentrations in 10/11 patients [151].

Most pediatric PK studies during CRRT have been performed with meropenem [97,98,137,138]. Three of the four studies investigating meropenem PK during CRRT included PICU patients, of which a subset received CRRT [97,98,137]. Two of these studies demonstrated a significant effect of CRRT on meropenem PK (i.e., CL or Vd) [97,98]. Rapp et al. showed that continuous infusion of meropenem was needed to achieve sufficient PTA during CRRT [97]. In the study of Saito et al., the central Vd increased by 66% in patients on CRRT [98]. In contrast, Wang et al. did not observe any significant PK alterations in patients receiving CRRT [137]. This might have been due to the fact that only 6/27 patients received CRRT, and that the sieving coefficient was much lower than previously reported for meropenem (0.26 vs. 0.95–1). Tan et al. performed the only meropenem PK study solely including children receiving CRRT [138]. They found that standard dosing regimens of meropenem lead to suboptimal target attainment. The authors recommended a longer infusion duration to reach sufficient PTA during CRRT. This study observed a mean sieving coefficient of 0.96, which is consistent with previous meropenem studies in adult patients. As the sieving coefficient approaches 1, this means that meropenem is quasi freely filtered during CRRT (which was applied as CVVH or CVVHDF in this study). While two studies did not find a significant influence of the effluent flow rate on meropenem CL in the subset of patients receiving CRRT [98,137], Tan et al. observed a significant association between the ultrafiltration and dialysate flow rate and meropenem CL [138]. This is consistent with an ex vivo study that showed increased CL of meropenem and piperacillin with increasing CRRT clearance rates in pediatric CRRT circuits [152].

**Table 4 antibiotics-10-01182-t004:** Overview of all antibiotic PK studies performed in critically ill children and infants with augmented renal clearance.

Reference	Antibiotic	Study Aspects	Age and Weight Range	Median (Range) or Mean ± SD eGFR (mL/min/1.73 m^2^)	Results and Clinical Implications
Giachetto et al. 2011 [52]	Vancomycin	Prospective study in mixed PICU patients with normal SCr (*n* = 22)	1 m–16 y Body weight not reported	Not reported	Standard dosing regimen 40–60 mg/kg/day in 4 doses infused over 1h leads to subtherapeutic exposure. Dosing recommendation: loading dose of 18–24 mg/kg needed in patients with fluid overload.
Silva et al. 2012 [56]	Vancomycin	Prospective popPK study in hemato-oncologic PICU patients (*n* = 31)	2 m–13 y 5–62 kg	136 ± 44.8 (mL/min)	Standard dosing regimen 40–60 mg/kg/day leads to subtherapeutic exposure
Gomez et al. 2013 [111]	Vancomycin	Prospective study in burns patients with sepsis and normal renal function (*n* = 13)	1–11 y 12–45 kg	221 (162–506) (mL/min)	Standard dosing regimen 40–45 mg/kg/day in 3–4 doses leads to subtherapeutic exposure Dosing recommendation: 90–100 mg/kg/day as initial dose in burns patients with sepsis and normal renal function
Hirai et al. 2016 [112]	Vancomycin	Retrospective popPK study in mixed PICU patients with eGFR > 90 mL/min/1.73 m^2^ (*n* = 109)	1–14.7 y 4.4–62.5 kg	160 (90–323)	Standard dosing regimen 40–60 mg/kg/day in 2–4 doses infused over 1h leads to subtherapeutic exposure in patients with febrile neutropenia Dosing recommendation: 60 mg/kg/day as initial dose in patients with febrile neutropenia (which were found to be at risk of ARC).
Avedissian et al. 2017 [64]	Vancomycin	Retrospective popPK study in mixed PICU patients with normal SCr (*n* = 250)	3.2–14 y (IQR) 15–50 kg (IQR)	Not reported	Standard dosing regimen 40–60 mg/kg/day in 3 doses leads to subtherapeutic exposure. Vancomycin CL was approx. 50 mL/min/1.73 m^2^ higher in children with ARC vs. those without ARC. Patients older than 7.9 years were more likely to experience ARC.
Lee et al. 2017 [113]	Vancomycin	Retrospective PK study in mixed PICU patients with eGFR ≥50 mL/min/1.73 m^2^ (*n* = 101)	3.7–13.6 y (IQR) 15–38.7 kg (IQR)	130 (91–163) (IQR)	Standard dosing regimen IQR 37–51 mg/kg/day in 3–4 doses leads to subtherapeutic exposure in 75% of the patients. An eGFR cutoff of 110.5 mL/min/1.73 m^2^ is predictive of subtherapeutic exposure.
Lv et al. 2020 [114]	Vancomycin	Retrospective popPK study with dosing simulations in patients with hematologic malignancy and eGFR ≥ 130 mL/min/1.73 m^2^ (*n* = 53, of which 15 PICU patients)	2.2–17.9 y 11–72 kg	258 (mean) (133–1284)	Standard dosing regimen leads to subtherapeutic exposure. Dosing recommendation: 50–75 mg/kg/day to achieve AUC/MIC ≥ 400. Higher dosage (per kg) needed with decreasing body weight.
He et al. 2021 [115]	Vancomycin	PopPK study with dosing simulations in infants, children and adolescents with ARC (*n* = 113)	0.4–14.9 y 6–62 kg	199 (160–332)	The currently recommended dosing regimen of 60 mg/kg/day leads to high risk of underdosing. Increased dosing is needed to reach the target AUC/MIC of 400–700: 75 mg/kg/day for infants and children (1 m–12 y), 70 mg/kg/day for adolescents (12–18 y).
Yu et al. 2015 [116]	Amikacin	Prospective popPK study with dosing simulations in burns patients with sepsis (*n* = 70)	2–10 y (IQR) 13–49 kg (IQR)	Not reported	Burns patients show markedly increased amikacin CL (and Vd) compared to patients without burns. Standard dosing regimens lead to subtherpaeutic exposure. Dosing recommendation: Increased doses (≥25 mg/kg) are needed to achieve C_max_/MIC ≥ 8.
Sridharan et al. 2020 [76]	Gentamicin	Retrospective popPK study in critically ill children (*n* = 73)	3.6 ± 4.6 (mean ± SD) 14.7 ± 16.5 (mean ± SD)	145 ± 87	Standard dosing regimens lead to lower AUC and subtherapeutic exposure in patients with ARC. Dosing recommendation: Monitor not only trough concentrations but also AUC in patients with ARC.
Cies et al. 2014 [84]	Piperacillin/tazobactam	Prospective popPK study with dosing simulations in mixed PICU patients (*n* = 13)	9 m–9 y 8.5–30 kg	Not reported	Standard dosing regimen 75–100 mg/kg q6h leads to subtherapeutic exposure in ARC patients Dosing recommendation: 400 mg/kg/day in 4 dose infused over 3h or in continuous infusion to achieve 50% *f*T_>MIC_.
Nichols et al. 2016 [81]	Piperacillin/tazobactam	Prospective popPK study with dosing simulations in mixed PICU patients with eGFR > 60 mL/min/1.73 m^2^ (*n* = 12)	1–9 y 9.5–30.1 kg	105 (86–189)	Standard dosing regimens lead to subtherapeutic exposure Dosing recommendation: 80–100 mg/kg q8h infused over 4 h to achieve 50% *f*T_>MIC_.
De Cock et al. 2017 [82]	Piperacillin/tazobactam	Prospective popPK study with dosing simulations in mixed PICU patients (*n* = 47)	2 m–15 y 3.4–45 kg	NA (many SCr concentrations below the limit of quantification)	Standard dosing regimen 75 mg/kg q6h infused over 5–30 min leads to subtherapeutic exposure in ARC patients. Dosing recommendation: 75 mg/kg q4h infused over 2h or 100 mg/kg q4h over 1h or loading dose of 75 mg/kg followed by continuous infusion of 300 mg/kg/day to achieve 60% *f*T_>MIC_.
Beranger et al. 2019 [83]	Piperacillin/tazobactam	Prospective popPK study with dosing simulations in PICU patients (*n* = 50)	0.1– 18 y 2.7–50 kg	142 (26–675)	Standard dosing regimens lead to subtherapeutic exposure Dosing recommendation: 400 mg/kg/day as extended or continuous infusion in patients with ARC to achieve 50% *f*T_>MIC_ or 100% *f*T_>MIC_.
De Cock et al. 2015 [93]	Amoxicillin/clavulanic acid	Prospective popPK study with dosing simulations in mixed PICU patients (*n* = 50)	1 m–15 y 4.07–65 kg	NA (many SCr concentrations below the limit of quantification)	Standard dosing regimens 25–35 mg/kg q6h infused over 5–30 min lead to subtherapeutic exposure. Dosing recommendation: 25 mg/kg q4h infused over 1h for ARC patients.
Cies et al. 2017 [95]	Meropenem	Retrospective popPK study with dosing simulations in mixed PICU patients with sepsis and eGFR >50 mL/min/1.73 m^2^ (*n* = 9)	1–9 y 7.5–40 kg	168 (104–224)	Standard dosing regimen 20–40 mg/kg q8h infused over 30 min leads to subtherapeutic exposure in ARC patients. Dosing recommendation: 20–40 mg/kg q6-8h infused over 3–4 h or continuous infusion to achieve 40% *f*T_>MIC_. 120–160 mg/kg/day as continuous infusion to achieve 80% *f*T_>MIC_.
Rapp et al. 2020 [97]	Meropenem	Prospective popPK study with dosing simulations in PICU patients (*n* = 40)	1.4 m–187.2 m 3.8–59 kg	151 (19–440)	Standard dosing regimens lead to subtherapeutic exposure Dosing advice: 60–120 mg/kg/day as continuous infusion to achieve 50% *f*T_>MIC_ or 100% *f*T_>MIC_.
Saito et al. 2021 [98]	Meropenem	Retrospective popPK study with dosing simulations in mixed PICU patients (*n* = 34)	0.03–14.6 y 2.7–40.9 kg	38.2 (1.4–183.8)	Standard dosing regimens lead to subtherapeutic exposure Dosing advice: 40–80 mg/kg q8h infused over 3h to achieve 100% *f*T_>MIC_.
Beranger et al. 2018 [87]	Cefotaxime	Prospective popPK study with dosing simulations in mixed PICU patients (*n* = 49)	6 d–19 y 2.5–68 kg	171 (40–304)	Standard dosing regimen 25–75 mg/kg q6h infused over 30 min leads to subtherapeutic exposure Dosing recommendation: 100 mg/kg/day as continuous infusion.
Cies et al. 2018 [91]	Ceftaroline	Retrospective popPK study in PICU patients with eGFR >60 mL/min/1.73 m^2^ (*n* = 7)	1–13 y 12.6–40.1 kg	Not reported	Standard dosing regimen leads to subtherapeutic exposure in ARC patients Dosing recommendation: 15 mg/kg q6h.

AUC: area under the curve; ARC: augmented renal clearance; CL: clearance; C_max_: peak concentration; eGFR: estimated glomerular filtration rate; *f*T_>MIC_: time during which the free concentration exceeds the minimum inhibitory concentration; IQR: interquartile range; MIC: minimum inhibitory concentration; NA: not available; PICU: pediatric intensive care unit; popPK: population pharmacokinetic; SCr: serum creatinine; SD: standard deviation; Vd: distribution volume.

**Table 5 antibiotics-10-01182-t005:** Overview of all antibiotic PK studies performed in critically ill children and neonates supported with extracorporeal membrane oxygenation.

Reference	Antibiotic	Study Aspects	Age and Weight Range	Results and Clinical Implications
Zylberstajn et al. 2018 [126]	Vancomycin	Retrospective popPK study in children <15 y (*n* = 40)	Age and body weight not reported for whole population (IQR varied between 1 m–100 m and 2.9–23 kg)	Standard dosing vancomycin 40–60 mg/kg/day in 4 doses. ECMO patients without acute kidney injury or RRT had similar Vd and lower CL lower than non-ECMO critically ill pediatric patients. Initial dosing must be adjusted according to renal function.
Amaker et al. 1996 [125]	Vancomycin	Prospective study in neonates (*n* = 12)	11 h–152 h 2.7–3.9 kg	Larger Vd and lower CL in ECMO vs. non-ECMO patients. Neonates without renal impairment should receive 20 mg/kg q24h. Estimated creatinine clearance is strongly associated with vancomycin CL, and can be used to guide vancomycin dosing.
Buck et al. 1998 [124]	Vancomycin	Retrospective study in neonates (*n* = 15 + 15 controls)	12.7 d ± 5.1 (mean ± SD) 3.1 kg ± 0.6 (mean ± SD)	Prolonged elimination but no statistically significant difference in CL and Vd compared to controls. Dosing interval should be extended beyond 6–8 h in neonates undergoing ECMO.
Cies et al. 2017 [121]	Vancomycin	Retrospective popPK study with dosing simulations in neonates with eGFR ≥10 mL/min/1.73 m^2^ (*n* = 12)	0–28 d 2.2–4 kg	More rapid CL compared to previous studies with older ECMO systems. More aggressive initial dosing regimen compared to older studies. A dosing interval q8-12h or continuous infusion is needed to achieve trough concentrations >10 mg/L.
Moffett et al. 2018 [123]	Vancomycin	Retrospective popPK study with dosing simulations in children <19 years without RRT (*n* = 93)	0.07–6.7 y (IQR) 3.7–21.9 kg (IQR)	SCr is associated with vancomycin CL. Infants and children showed lower target attainment, and had lower SCr concentrations. 25–30 mg/kg q12-24h had the greatest likelihood of achieving targt AUC/MIC values. Higher doses might be needed for patients with normal renal function. Likewise, increased dosing intervals might be necessary for patients with increased SCr concentrations.
An et al. 2019 [122]	Vancomycin	Prospective study in neonates (*n* = 25 + 25 controls)	8 ± 7.9 (mean ± SD) 3.1 ± 0.4 (mean ± SD)	Increased elimination half-life and decreased CL in the ECMO patients compared to the controls. Vd was unchanged. CL was associated with SCr. However ECMO remained significantly associated with CL after adjusting for SCr. It is unclear if analyses were also adjusted for RRT. Dosing adjustments and TDM are required in neonates undergoing ECMO.
Southgate et al. 1989 [127]	Gentamicin	Prospective study in neonates (*n* = 10)	Post-natal age not reported (‘soon after birth’) 2.7–4.8 kg	Similar Vd to non-ECMO neonates. Longer elimination half-life compared to non-ECMO neonates. SCr was strongly associated with the elimination half-life. A dose of 2.5 mg/kg is a reasonable starting point. The dosing interval varied between 8–30 h and needs to be determined by TDM.
Cohen et al. 1990 [128]	Gentamicin	Prospective popPK study in neonates (*n* = 18, of which 12 were their own control after ECMO discontinuation)	2–8 d Body weight not reported	Increased Vd in ECMO vs. non-ECMO patients. CL was decreased by 25% due to a decrease in eGFR. A 25% increase in dose and longer dosing intervals should be considered.
Munzenberger et al. 1991 [129]	Gentamicin	Retrospective study in neonates (*n* = 15)	Age not reported 2.5–6.6 kg	No major impact of ECMO on gentamicin Vd and CL when compared to previous studies in neonates not supported with ECMO.
Bhatt-Mehta et al. 1992 [130]	Gentamicin	Retrospective study in neonates (*n* = 29)	Post-natal age not reported 3.35 ± 0.71 kg (mean ± SD)	Similar Vd and prolonged elimination half-life in ECMO patients compared to previously reported Vd in non-ECMO neonates. No significant differences in Vd and CL between different types of ECMO (veno-arterial vs. veno-venous) and oxygenator. A dosing regimen of 2.5 mg/kg q18h is recommended for neonates undergoing ECMO. TDM should be used to individualise the dosing interval.
Dodge et al. 1994 [131]	Gentamicin	Prospective popPK study in neonates (*n* = 11, of which 6 were their own control after ECMO discontinuation)	Post-natal age not reported 2.7–5.1 kg	Decrease in Vd and increase in CL after discontinuation of ECMO. Increase in loading dose (>2.5 mg/kg) and prolonged dosing interval beyond 8–12h is needed.
Moffett et al. 2018 [132]	Gentamicin	Retrospective popPK study with dosing simulations in children <19 y (*n* = 37)	0.12–0.82 m (IQR) 2.4–3.8 kg (IQR)	Fat-free mass should be used to guide gentamicin dosing. SCr is predictive of gentamicin CL, and should be used to adjust dosing. Dosing regimens for neonates and infants undergoing ECMO are not significantly different from those in non-ECMO patients. A dosing regimen of 4–5 mg/kg q24h is acceptable for neonates and infants. Older children may need longer dosing intervals to compensate for increased trough concentrations.
Cies et al. 2011 [136]	Piperacillin/tazobactam	Retrospective popPK study with dosing simulations (*n* = 6)	8 d–7 m Body weight not reported	Trough concentrations of 150 mg/L were not attained. A dosing regimen of 50 mg/kg q2h, 100 mg/kg q4h or 200 mg/kg q6g would be needed to obtain a piperacillin trough concentration of 150 mg/L.
Lindsay et al. 1996 [141]	Ticarcillin/clavulanic acid	Prospective study (*n* = 3, of which 2 patients were on ECMO)	6–6.5 y 22.1–27.6 kg	CL of ticarcillin and clavulanic acid were comparable in the two patients receiving ECMO and the third patient not receiving ECMO (all patients received CRRT). CL is decreased compared to historical controls.
Cies et al. 2014 [153]	Meropenem	Case report in a pediatric cardiac critically ill patient (*n* = 1)	8 m Body weight not reported	Meropenem 40 mg/kg bolus followed by continuous infusion of 200 mg/kg/day resulted in attainment of 40% *f*T_>MIC_ in serum and lungs, and was associated with successful clinical outcome in a patient infected with *P. aeruginosa* with a MIC of 0.5 mg/L.
Alqaqaa et al. 2016 [140]	Meropenem	Case report in a pediatric patient with septic shock supported with ECMO and CRRT	4 y	Significantly reduced CL compared with other case reports. Supratherapeutic concentrations were obtained with continuous infusion of 10 mg/kg/h (following an intermittent dosing of 40 mg/kg q8h).
Cies et al. 2016 [139]	Meropenem	Case report in a full-term neonate with concomitant ECMO and CCRT (*n* = 1)	10 d 2.8 kg	CL is larger than in a previous case report with ECMO but without CRRT. Meropenem 40 mg/kg bolus followed by continuous infusion of 240 mg/kg/day resulted in attainment of 40% *f*T_>MIC_ in serum, and was associated with microbiological eradication in a patient infected with *P. aeruginosa* with a MIC of 0.25 mg/L.
Rapp et al. 2020 [97]	Meropenem	Prospective popPK study with dosing simulations in PICU patients (*n* = 40, of which 8 received ECMO)	1.4 m–187.2 m 3.8–59 kg	ECMO had no significant effect on meropenem Vd and CL.
Saito et al. 2021 [98]	Meropenem	Retrospective popPK study with dosing simulations in mixed PICU patients (*n* = 34, of which 3 received ECMO)	0.03–14.6 y 2.7–40.9 kg	ECMO had no significant effect on meropenem Vd and CL.
Tan et al. 2021 [138]	Meropenem	Prospective popPK study with dosing simulations in children receiving CRRT (*n* = 9, of which 4 received ECMO)	0.1–18.9 y 2.6–56.3 kg	ECMO had no significant effect on meropenem Vd and CL.
Wang et al. 2021 [137]	Meropenem	Propspective study in children with sepsis (*n* = 27, of which 6 received ECMO)	0.5–5.2 y (IQR) 6.5–21.5 kg (IQR)	ECMO did not significantly alter meropenem Vd and CL, hence dosing adjustments might not be needed in patients supported with ECMO.
Ahsman et al. 2010 [134]	Cefotaxime	Prospective popPK study with dosing simulations (*n* = 37)	1–199 d 2–6.2 kg	Larger Vd and comparable CL compared to non-ECMO patients. The standard dosing regimen (50 mg/kg q12h for PNA <1 week; 50 mg/kg q8g for PNA 1–4 weeks; 37.5 mg/kg q6h for PNA >4 weeks) provides sufficient exposure (50% *f*T_>MIC_) in infants receiving ECMO.
Zuppa et al. 2019 [135]	Cefepime	Prospective popPK study (*n* = 17)	1.4–22.2 m 3.3–10 kg	CL was reduced and Vd increased compared to non-ECMO children. The central Vd decreased with increasing age of the ECMO oxygenator. Only 14/19 doses studied achieved 70% *f*T_>MIC_.
Yang et al. 2021 [102]	Linezolid	Prospective popPK study with dosing simulations (*n* = 63, of which 2 received ECMO)	0.1–15.3 y 4.2–70 kg	ECMO had no significant effect on linezolid Vd and CL.

AUC: area under the curve; (C)RRT: (continuous) renal replacement therapy; CL: clearance; ECMO: extracorporeal membrane oxygenation; eGFR: estimated glomerular filtration rate; *f*T_>MIC_: time during which the free concentration exceeds the minimum inhibitory concentration; IQR: interquartile range; MIC: minimum inhibitory concentration; PICU: pediatric intensive care unit; PNA: postnatal age; popPK: population pharmacokinetic; SCr: serum creatinine; SD: standard deviation; TDM: therapeutic drug monitoring; Vd: distribution volume.

**Table 6 antibiotics-10-01182-t006:** Overview of all antibiotic PK studies performed in critically ill children and neonates supported with cardiopulmonary bypass.

Reference	Antibiotic	Study Aspects	Age and Weight Range	Results and Clinical Implications
Haessler et al. 2003 [144]	Cefazolin, Gentamicin	Prospective study in children less than 10 kg during and after CPB (*n* = 19)	1 d–2.6 y 3.8–10.5 kg	Increase in Vd during CPB for cefazolin and gentamicin. Vd returned to baseline for cefazolin. Vd increased further after surgery for gentamicin. The elimination constant decreased during surgery for both antibiotics. However, it increased after surgery for cefazolin, but remained low after surgery for gentamicin. For cefazolin, 40 mg/kg at induction of anesthesia, followed by 35 mg/kg q8h for 48 h lead to sufficiently high trough concentrations (>8 mg/L). For gentamicin, 5 mg/kg at induction of anesthesia, followed by 2 mg/kg q12h for 48 h lead to unnecessary high trough concentrations (>6–8 mg/L). Therefore, a reduced dose of 2 mg/kg, with a second dose after 4 h if surgery is not yet completed, and no postoperative dose, is recommended
Cies et al. 2019 [154]	Cefazolin	Prospective popPK study in children during CPB (*n* = 41)	From birth to 16 y (range not specified) 3.5–79 kg (IQR for lowest and highest age group, respectively)	Mixing cefazolin (25 mg/kg) in the CPB priming fluid resulted in adequate exposure during cardiac surgery. Repeated administration of cefazolin during and after surgery is probably not needed.
De Cock et al. 2017 [142]	Cefazolin	Prospective popPK study with dosing simulations in children before, during and after CPB (*n* = 56)	6 d–15 y 2.7–70 kg	Increase in Vd during CPB. Decreasing eGFR during and after CPB leads to decreased CL. Subtherapeutic exposure in a substantial fraction of patients, especially in patients with prolonged cardiac surgery and preserved renal function. An adapted dosing regimen consisting of: 40 mg/kg, 30 min before surgical incision; 20 mg/kg, at start of CPB and at start of rewarming on CPB; and 40 mg/kg 8 h after the third and fourth dose improves the probability of target attainment from 40% to >88% for infections caused by staphylococci.
Gertler et al. 2018 [143]	Cefuroxime	Prospective popPK study with dosing simulations in children less than one year with congenital heart defect undergoing cardiac surgery with (*n* = 36) or without (*n* = 6) CPB	6–348 d 2.3–9.5 kg	No effect on Vd, but significantly increased CL during CPB. An adapted dosing regimen of 25 mg/kg bolus followed by continuous infusion of 5 mg/kg/h, with an additional 25 mg/kg bolus in the priming fluid maintains free plasma cefuroxime concentrations above 4xMIC (i.e., 32 mg/L). If continuous infusion is not used, intermittent infusion of 50 mg/kg q2h is recommended.

CPB: cardiopulmonary bypass, CL: clearance; eGFR: estimated glomerular filtration rate; IQR: interquartile range; Vd: distribtuion volume.

**Table 7 antibiotics-10-01182-t007:** Overview of all antibiotic PK studies performed in critically ill children and neonates supported with continuous renal replacement therapy.

Reference	Antibiotic	Study Aspects	Age and Weight Range	Results and Clinical Implications
Cies et al. 2016 [151]	Vancomycin	Retrospective study in children receiving CRRT (CVVH, CVVHD and CVVHDF) (*n* = 11)	0.08–18 y 3.1–65 kg	Mixing of vancomycin in the CRRT solution lead to therapeutic plateau concentrations (15–30 mg/L) in 10/11 patients 8 h after start of CRRT.
Moffett et al. 2019 [150]	Vancomycin	Retrospective popPK study with dosing simulations in children receiving CVVHDF (*n* = 138)	1–14.5 y (IQR) 31 ± 25.8 kg (mean ± SD)	The CL is within the range of values observed in non-CVVHDF children. The ultrafiltration rate and dialysis rate have a significant positive effect on the CL. There is high variability in dosing regimens needed to attain therapeutic exposure. The most optimal empiric dosing regimen is 15 mg/kg q8h based on fat-free mass, which leads to target attainment in approx. 80% of patients.
Girdwood et al. 2020 [155]	Piperacilllin/tazobactam	Case report in one child with sepsis and liver failure supported with concomitant CRRT and MARS	13 y 42 kg	50% *f*T_>4xMIC_ was attained under CRRT alone, but not during MARS due to increased CL of piperacillin. Higher dose and extended infusion are recommended when initiating MARS, but not for CRRT alone.
Rapp et al. 2020 [97]	Meropenem	Prospective popPK study with dosing simulations in PICU patients (*n* = 40, of which 11 received CRRT)	1.4 m–187.2 m 3.8–59 kg	Standard dosing regimens lead to subtherapeutic exposure not only in patients with normal or augmented renal clearance, but also in CRRT patients. CRRT had a significant effect on CL. A dosing regimen of 60 mg/kg/day as continuous infusion is needed to achieve 50% *f*T_>MIC_ or 100% *f*T_>MIC_. In case of MIC values <2 mg/L, intermittent infusion of 20 mg/kg q8h is also adequate.
Saito et al. 2021 [98]	Meropenem	Retrospective popPK study with dosing simulations in mixed PICU patients (*n* = 34, of which 8 received CRRT [CVVHDF and CVVHD])	0.03–14.6 y 2.7–40.9 kg	Standard dosing regimens lead to suboptimal exposure. Vd was 66% higher in CRRT patients. A dosing regimen of 40–80 mg/kg q8h infused over 3 h is needed to achieve 100% *f*T_>MIC_.
Tan et al. 2021 [138]	Meropenem	Prospective popPK study with dosing simulations in children receiving CRRT (CVVH or CVVHDF) (*n* = 9)	0.1–18.9 y 2.6–56.3 kg	Meropenem is freely filtered over the hemofilter/dialysis membrane (mean sieving coefficient of approx. 1). Standard dosing regimen of 40 mg/kg q12h lead to suboptimal exposure when targeting 100% *f*T_>MIC_ in CRRT patients. 20 mg/kg q8h over 4 h or 40 mg/kg q8h over 2 h are needed to achieve 100% *f*T_>MIC_.
Wang et al. 2021 [137]	Meropenem	Prospective study in children with sepsis (*n* = 27, of which 6 patients received CVVHDF)	1.13–6.88 y (IQR) 8.6–27.1 kg (IQR)	Lower sieving coefficient than previously reported (0.26 vs. approx. 1). CRRT intensity (no high-flow filtration was used in this study) did not significantly alter meropenem Vd and CL, hence dosing adjustments might not be needed in patients supported with CRRT.
Stitt et al. 2019 [156]	Cefepime	Retrospective study in children receiving CVVHDF (*n* = 4)	0.5–5 y 5.4–25 kg	The standard dosing regimen of 50 mg/kg q12h might not be sufficient to reach 100% *f*T_>1–4xMIC_. The CVVHDF clearance might be the driver of decreased exposure.
Butragueño-Laiseca et al. 2020 [90]	Ceftolozane/tazobactam	Case series with one patient receiving CVVHDF and two patients without CRRT	8–19 m 5.8–11 kg	The CL was approx. half that of the patient with normal renal function. A dosing regimen of 30 mg/kg q8h resulted in adequate target attainment.
Yang et al. 2021 [102]	Linezolid	Prospective popPK study with dosing simulations (*n* = 63, of which 15 received CRRT)	0.1–15.3 y 4.2–70 kg	CRRT had no significant effect on linezolid Vd and CL.

CL: clearance; CRRT: continuous renal replacement therapy; CVVH: continuous veno-venous hemofiltration; CVVHD: continuous veno-venous hemodialysis; CVVHDF: continuous veno-venous hemodiafiltration; *f*T_>(4×)MIC_: time during which the free concentration exceeds (four times) the minimum inhibitory concentration; IQR: interquartile range; MARS: molecular adsorbent recirculating system therapy; MIC: minimum inhibitory concentration; PICU: pediatric intensive care unit; popPK: population pharmacokinetic; SD: standard deviation; Vd: distribution volume.

One case report described the PK of piperacillin during CRRT and molecular adsorbent recirculating system (MARS) therapy [155]. Piperacillin/tazobactam 48 mg/kg q8h as an intermittent infusion over 30 min led to adequate PK/PD target attainment during CRRT alone. However, piperacillin CL more than doubled during MARS, and exposure decreased below the PK/PD target. Between MARS cycles, CL decreased back to approximately the initial values. Consequently, the authors conclude that subtherapeutic piperacillin exposure is rather due to MARS than to CRRT.

One pediatric ICU study investigating linezolid PK included a subset of children with CRRT [102]. CRRT was not retained in their final popPK model. Therefore, CRRT did not appear to influence the PK of linezolid.

For cefepime, a case series of 4 patients receiving CVVHDF showed that standard 50 mg/kg q12h might not be sufficient to reach predefined PK/PD targets [156]. Dosing may need to be increased depending on the CVVHDF clearance rate, and TDM was recommended by the authors.

Finally, a recent case series of three critically ill children with different renal functions (normal, decreased, and CRRT) treated with ceftolozane/tazobactam described the PK differences amongst these three patients [90]. The CL of ceftolozane/tazobactam under CRRT was approximately half of the CL in the child with normal renal function, which has also been described in adult patients. The sieving coefficient was near 1 (0.99–1.14), which was consistent with values previously reported in adult patients treated with continuous veno-venous hemofiltration and hemodialysis. The authors stated that 30 mg/kg q8h should lead to adequate target attainment in children receiving CRRT with high effluent flow rate.

Overall, in children treated with CRRT, antibiotic PK, and consequently antibiotic dosing, appears to be dictated by the effluent flow rate (ultrafiltration ± dialysis flow rate).

### 4.3. Whole Body Hypothermia in Neonates

WBH has a neuroprotective effect in (near)term neonates with moderate to severe hypoxic ischemic hypothermia and is applied to reach a target temperature of 33–34 °C for 72 h shortly (<6 h) after birth. However, hypoxic ischemic asphyxia and lowering the core body temperature affects PK, up to the level that dosing regimens for this specific setting should be adapted. A significant decrease in CL is observed in neonates during therapeutic hypothermia, be it that the extent differs between routes of elimination and compounds, but is most pronounced for renal elimination [157,158]. Based on a systematic review on SCr trends and patterns in neonates undergoing WBH, the impact on renal elimination is most relevant in the first days of life, so during WBH (used in the first 3 days of postnatal life) [159]. Consequently, this time window overlaps with the ‘early onset sepsis (EOS)’ interval, and can affect PK and dosing of the most commonly used antibiotics in the EOS setting, like aminoglycosides or beta lactams, as reflected in Table 8 [160].

Table 8 provides an overview of the impact of WBH on the PK of gentamicin, amikacin, ampicillin, amoxicillin, and benzylpenicillin. Based on PK observations in WBH cases, the general pattern reflects a relevant (−20 to −40%) decrease in renal elimination, without relevant effects on distribution. As such, dosing can be adapted by either extending the time interval between consecutive doses (e.g., +q12 h) for aminoglycosides or reducing the dose for beta-lactams.

## 5. Discussion

Based on the pediatric study decision tree, the maturational physiology throughout pediatric life results in more complex antibiotic dosing regimens to attain similar target exposure in children as compared to adults [19,20]. As highlighted in this paper, the maturational variability in antibiotic PK is further affected by pathophysiological changes, like critical illness. This variability—if handled inaccurately—will result in either under- or overexposure in a significant proportion of patients. Consequently, dose selection and tools to support dose decisions are crucial to improving pharmacotherapy of antibiotics in children to attain precision medicine (Figure 3).

In a first step (Figure 3, best guess), selection of the most appropriate dosing regimen, applicable to the individual patient is relevant, as not all dosing regimens have been developed in similar populations. More adapted dosing regimens can take several covariates (like age, weight, SCr, co-morbidity, or co-medication) into account in a single nomogram. The prospective validation of an amikacin nomogram in (pre)term neonates hereby illustrated that such an approach resulted in better exposure and target attainment (peak, trough) [174]. Nowadays, population PK modelling is often used to suggest optimal dosing regimens for antibiotics in special patient populations—such as critically ill patients. Population PK modelling allows the estimation of PK parameters (e.g., CL and Vd) at a population level. Based on a population PK model, predictions can be made about the future by simulating drug exposure [175,176]. As such, drug exposure can be predicted according to different dosing regimens. One of the crucial criteria hereby is the presence of an external prospective validation to select the most appropriate dosing regimen for the intended population, while individual dosing selection can be supported by MIPD programs. In a systematic literature search on population PK of antibiotics with predominant renal elimination in neonates by Wilbaux et al., external validation was only reported in 20% of the studies retained in their search [13]. To further illustrate the relevance of applicability, omissions, or specific exclusion criteria, e.g., cases with severe renal impairment that are frequently excluded during the development of the initial model, will obviously result in a model and dosing regimen not applicable in these cases during subsequent clinical use. This has been recently illustrated by Hartman et al. in an external validation of model-based dosing guidelines for vancomycin, gentamicin, and tobramycin in critically ill neonates and children: failed efficacy target attainment for vancomycin (66%) and gentamicin (69%), and failed safety target attainment for gentamicin (15%) or tobramycin (23%) were commonly observed [177]. We would like to add that besides population characteristics, some technical aspects—like the type of SCr or vancomycin assays—may also affect the transferability of published models to different clinical settings [178].

For the second step (Figure 3, tailor to target)—historically—TDM at steady state-assisted health care providers to assess target attainment and to adapt the dosing regimen if needed. However, one of the basic rules on TDM usefulness mentions that the PK and its covariates are sufficiently well known so that a TDM observation can be used to adapt the treatment. Unfortunately, the variability in PK and patient covariates within the context of critical illness makes an accurate prediction of the dosing regimen or dose adaptation needed for optimal exposure at the individual level complex when using only the TDM result. MIPD might overcome this issue, and provide more accurate predictions for certain antibiotics in certain patient populations. As it combines TDM with population PK modeling, MIPD is a valuable dosing strategy, next to TDM alone, to optimise individual dosing regimens of antibiotics. MIPD involves the use of popPK models and prospective Bayesian forecasting to reduce variability in response (i.e., PD). A Bayesian approach delivers a population-estimated value for each PK parameter, including the variability components, i.e., variability due to real biological differences between individuals (inter-individual variability) as well as unexplained variability (noise, residual error) and simultaneously [18].

Our review rather intended to summarise the available evidence on (non-)maturational covariates in the NICU and PICU setting. The next step to convert this information to attain precision medicine, based on Bayesian and MIPD software techniques, has been recently reviewed by others so that we refer the interested reader to this review [18].

Han et al. recently identified and assessed the Bayesian software programs applicable to neonatal and pediatric patients for vancomycin (AUC over 24 h exposure) in such an MIPD context. Of the Bayesian programs retrieved, DoseMeRx, InsightRx, and PrecisePK utilised clinically validated neonatal and pediatric models. The authors came to the conclusion that clinicians should focus on selecting a model that best fits their patient population characteristics and utilise Bayesian estimation tools for therapeutic AUC-targeted dosing and monitoring of vancomycin [179].

Finally, the third step (Figure 3, confirm target) is crucial in the setting of hyperdynamic and fast maturational and non-maturational changes (likely even more dynamic in an ICU setting) in distribution and elimination, such as in critically ill neonates and children. Sustained targeted exposure should not be taken for granted because of these fast changes in individual PK characteristics. Consequently, this necessitates the continuation of the TDM + MIPD practices discussed in the second step.

Obviously, such an approach necessitates additional multidisciplinary collaboration as the skills related to MIPD should be integrated into clinical care, while further evidence on efficacy and cost-benefit analysis of MIPD in healthcare is needed before MIPD will become common clinical practice [180]. Furthermore, as evidenced by the studies identified in this review, TDM is mainly performed for vancomycin and aminoglycosides. However, TDM services should perhaps be extended to other much less commonly measured antibiotics, like beta-lactams [181]. Although beta-lactam TDM is on the rise, recent national surveys revealed that beta-lactam TDM is still not common clinical practice in most ICUs [182,183]. This is due to several practical challenges, of which the lack of a commercially available beta-lactam assay is probably the most important [175]. Finally, we should realise that improved target attainment is only (a likely relevant) surrogate marker for clinical outcomes. At present, there are only a few observations on the impact of MIPD for antibiotics on target attainment. In critically ill children, Hughes et al. documented that 70.8% of children with MIPD-guided vancomycin dosing attained the trough concentration target, whereas only 37.5% (54/144) of children in the clinician-guided arm attained these targets [184]. Leroux et al. showed not only improved target attainment with MIPD-guided dosing in neonates (41% in the standard regimen vs. 72% in the MIPD arm), but they also observed a lower incidence of nephrotoxicity (8.7% vs. 1.1%) in the MIPD arm [185]. Smits et al. published the only MIPD study performed with another antibiotic than vancomycin, i.e., amikacin. Their model-based approach resulted in optimal exposure (i.e., peak and trough concentrations within the target range) in almost all neonates [174]. Still, none of these studies investigated a clinical primary endpoint, nor was the cost-effectiveness of an MIPD strategy evaluated. In adults, several retrospective or small clinical trials have suggested a clinical benefit from antibiotic TDM, with or without MIPD [186]. Recently, preliminary results of the TARGET trial, a large multicenter randomised controlled trial (RCT) in 10 German ICUs, failed to show a clinical benefit from TDM-guided dosing of piperacillin-tazobactam in critically ill adults [187]. Interestingly, TDM-guided dosing resulted in better target attainment, and under- or overexposure was associated with higher mortality rates. It might be that TDM-guided dosing did not lead to better clinical outcomes due to the fact that a significant proportion of patients still showed under- or overexposure in the intervention arm (approx. 50%). MIPD, with TDM, might further increase target attainment and eventually lead to better clinical outcomes. Meanwhile, evidence to confirm the potential clinical benefit of MIPD for antibiotics is still under investigation in the DOLPHIN trial, a multicenter RCT that evaluates the effect of beta-lactam and fluoroquinolone MIPD on the ICU length of stay. Once available, these data will likely be extrapolated—applying the previously discussed concepts [19,20]—to the pediatric field [188].

## Figures and Tables

**Figure 1 antibiotics-10-01182-f001:**
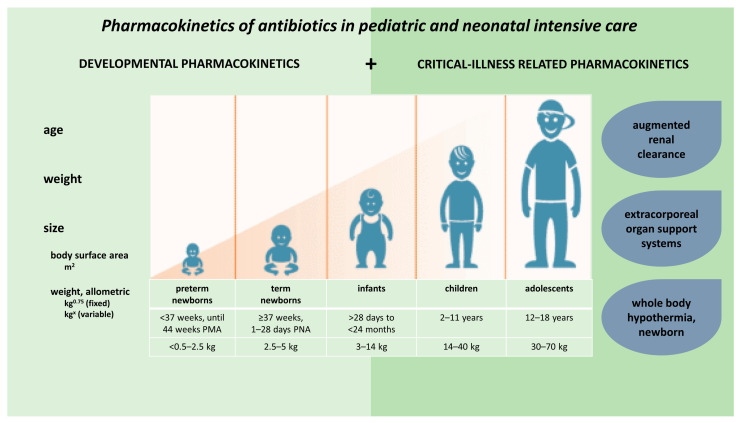
Illustration of developmental and critical-illness related pharmacokinetic alterations in children and neonates.

**Figure 2 antibiotics-10-01182-f002:**
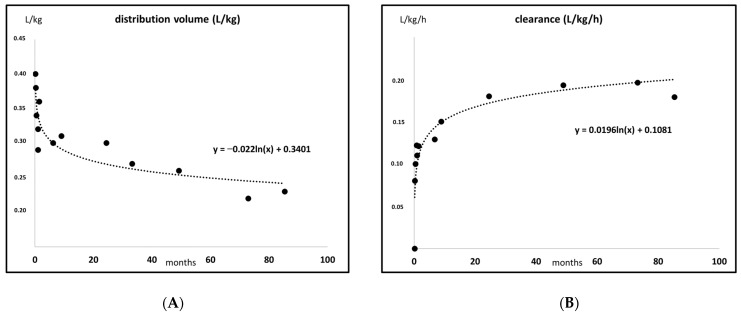
Illustration of maturational changes of (**A**) volume of distribution (Vd, L/kg) and (**B**) clearance (CL, L/kg/h) with the trend lines, based on amikacin pharmacokinetic data in 155 patients (body weight 1.35–33 kg) in the first 85 months of life [28].

**Figure 3 antibiotics-10-01182-f003:**
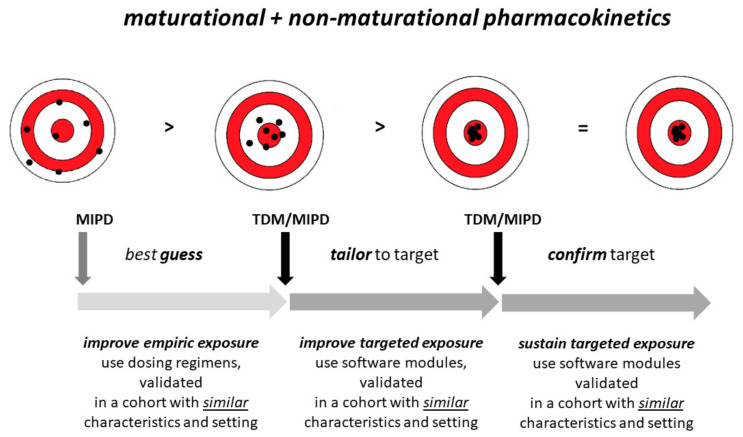
Structured approach to improve target exposure for antibiotics in critically ill neonates and children.

**Table 1 antibiotics-10-01182-t001:** Maturational changes in physiologic variables relevant to the distribution of antibiotics [25,26,27].

Age	Total Body Water/Weight %	Extracellular Water/Weight %	Intracellular Water/Weight %	Plasma Albumin g/L	Plasma Total Protein d/L
Preterm, 2 kg	82	44	34	26	40
Term, at birth	78	40	32	28	43
7–30 days	74	32	43	29	53
1–3 months	73	30	42	29	54
3–6 months	70	30	42	29	54
6–12 months	60	27	35	29	54
1–3 year	58–63	27–21	38–34	31	59
>3–6 year	62	21	46	31	62
>6–18 year	61–57	22–18	43–42	32	59
Adult	59	19	40	40	63

**Table 2 antibiotics-10-01182-t002:** Categories of antibiotics according to their pattern of antibacterial activity and their associated pharmacokinetic/pharmacodynamic (PK/PD) target.

Category Based on the In Vitro Pattern of Antibacterial Activity	Antibiotic Class	PK/PD Target for Efficacy
Time-dependent with minimal-to-no persistent effect	Beta-lactams, Lincosamides	% *f*T_>MIC_
Time-dependent with moderate-to-persistent effect	Glycopeptides, Oxazolidinones, Macrolides, Tetracyclines, Glycylcyclines, Polymyxins	AUC_0–24_/MIC % *f*T_>MIC_ (erythromycin, linezolid)
Concentration-dependent with moderate-to-persistent effect	Aminoglycosides, Fluoroquinolones, Lipopeptides, Metronidazole, Ketolides	AUC_0–24_/MIC C_max_/MIC

*% f*T_>MIC_: percentage of time during which the free concentration exceeds the minimum inhibitory concentration; AUC_0–24_/MIC: ratio of the area under the curve over 24 h to the minimum inhibitory concentration; C_max_/MIC: ratio of the peak concentration to the minimum inhibitory concentration.

**Table 3 antibiotics-10-01182-t003:** Summary of all antibiotic PK studies performed in critically ill children.

Antibiotic	Number of Studies	Study Aspects	Age and Weight Range	Results and Clinical Implications
Vancomycin	19	4 prospective studies (*n* = 168) [50,51,52,53] 10 retrospective TDM studies (*n* = 1120) [54,55,56,57,58,59,60,61,62,63] 5 retrospective popPK studies (*n* = 704) [64,65,66,67,68]	0–18 y 0.68–108 kg	Conflicting results with regard to association between C_min_ and acute kidney injury [54,55,57,62] Eight studies reported CL and Vd. Mean Vd 0.44–1.04 L/kg. Mean CL 0.072–0.19 L/kg/h [51,52,53,56,64,65,67,68] Eight studies reported measured, simulated or estimated AUCs [50,51,52,56,62,66,67,68] Only two studies used continuous dosing [50,66] Substantial percentage of target non-attainment with standard dosing regimens (up to 92%, mostly subtherapeutic but also supratherapeutic concentrations) [50,51,52,54,55,56,59,64,66,67,68] Dosing of 60 mg/kg/day q8h advised if no renal impairment [61,67] One study advised lower doses for neonates (30 mg/kg/day), infants (35–40 mg/kg/day) and children (45 mg/kg/day) [59] Another study in neonates and infants <2m advised 14–18 mg/kg q8–12 h [68]
Teicoplanin	4	2 prospective cohort studies (*n* = 33) [69,70] 1 RCT (*n* = 20) [71] 1 prospective popPK study with dosing simulations (*n* = 42) [72]	7 d–15.6 y 3.74–56 kg	Three studies found that higher than standard dosing is needed to achieve Target attainment [70,71,72] One study found lower target attainment in older children (>1 y) compared to younger infants (<1 y) due to larger Vd and higher CL [71] Routine TDM of unbound concentrations was recommended due to highly variable unbound concentrations [72]
Gentamicin	4	1 retrospective TDM study (*n* = 140) [73] 1 prospective popPK study with dosing simulations (*n* = 36) [74] 2 studies investigating application of a Bayesian forecasting program (*n* = 117) [75,76]	0 d–15 y Body weight not reported	Higher initial doses and/or extended dosing interval in neonates and (young) infants [74,75] Two studies found age and weight to be significant predictors for Vd and/or CL [73,74] One study also found serum creatinine to be a significant predictor for the elimination constant (k) [73]
Amikacin	3	1 RCT (*n* = 60) [77] 2 prospective popPK studies (*n* = 106) [78,79]	6 m–17 y 8–90 kg	Higher doses per kg needed for neonates and infants (<1 y) due to higher Vd and CL [77] Higher Vd and CL in burn patients [79]
Netilmicin	1	1 prospective study (*n* = 9) [80]	1 m–15.5 y 3.4–70 kg	Mainly neonates Once daily 6 mg/kg is sufficient
Piperacillin/tazobactam	5	4 popPK studies with dosing simulations (*n* = 139) [81,82,83,84] 1 prospective study (*n* = 14) [46]	0.1–18 y 2.7–53 kg	High median eGFR_Schwartz_ in all studies (lowest median eGFR_Schwartz_ 142 mL/min/1.73 m^2^) [83] Median Vd: 0.24–0.444 L/kg (highest in neonates); Median CL: 0.19–0.299 L/kg/h [81,82,83,84] Insufficient target attainment with standard dosing. Extended infusion over >1 h needed for >90% probability of target attainment [46,81,82,83,84]
Cefotaxime	3	2 prospective studies (*n* = 39) [85,86] 1 prospective popPK study with dosing simulations (*n* = 49) [87]	0–19 y 2.5–70 kg	Neonates had longer elimination half-life [87] Continuous infusion needed for optimal target attainment and/or less susceptible microorganisms [85,87]
Cefuroxime	1	1 prospective cohort study (*n* = 11) [88]	4 m–14 y 5.1 kg–45 kg	Vd and CL higher in children with mechanical ventilation vs. children without mechanical ventilation and controls The elimination half-life is longer in critically ill children vs. controls
Ceftriaxone	1	Prospective popPK study with dosing simulations (*n* = 45) [89]	0.1–16.7y	Vd and CL comparable to non-critically ill children aged 1–6 y Vd and CL higher than non-critically ill children with cystic fibrosis 100 mg/kg q24h sufficient for most critically ill children and neonates 50 mg/kg q12h if eGFR_Schwartz_ > 80 mL/min/1.73 m^2^ or increased MIC ≥0.5 mg/L
Ceftolozane/tazobactam	1	1 case series (*n* = 3) [90]	8–19 m 5.8–11 kg	Normal renal function: 35 mg/kg q8h appropriate for multidrug-resistent *Pseudomonas aeruginosa* Acute kidney injury: reduced dose 10 mg/kg q8h appropriate
Ceftaroline	1	1 prospective study (*n* = 7) [91]	1–13 y 12.6–40.1 kg	Higher median CL and Vd than reported in the package insert (non-critically ill children) Higher dosing and shorter dosing interval than package insert needed (15 mg/kg q6h)
Amoxicillin/clavulanic acid	3	1 prospective study (*n* = 15) [92] 1 prospective popPK study with dosing simulations (*n* = 50) [93] 1 meta-analytical modelling study (*n* = 44) [94]	1 d–15 y 1.7–65 kg	Higher amoxicillin CL than critically ill adults, comparable amoxicillin Vd and clavulanic acid CL and Vd [92,93] 25 mg/kg q4h as bolus or 1h infusion, depending on renal function, needed for optimal target attainment [93] Meta-modelling study (in neonates and young infants (<60 d) [94]: Sepsis is associated with lower amoxicillin concentrations and longer elimination half-life. Fixed dosing regimen: 125 mg and 250 mg q12h depending on body weight <4 kg or ≥4 kg
Meropenem	5	1 retrospective popPK study (*n* = 9) [95] 1 case report (*n* = 1) [96] 1 prospective popPK study with dosing simulations (*n* = 23) [97] 1 retrospective popPK study with dosing simulations (*n* = 26) [98] 1 prospective study in children with sepsis (*n* = 15) [99]	0.03–15.6 y 2.7–59 kg	CL is slightly lower [99], within [97], or higher [95,96,98] than the CL range observed in non-critically ill children, depending on the study population Increased dosing and extended infusion needed [95,96,97,98,99]
Imipenem	1	1 prospective study (*n* = 19) [100]	9 d–12 y Body weight not reported	Vd and CL comparable to non-critically ill children At least 100 mg/kg/day to avoid subtherapeutic concentrations
Aztreonam	1	1 case report (*n* = 1) [101]	16 y	CL double than reported in the package insert 2g q6h over 4h infusion achieved 40% *f*T_>MIC_
Linezolid	1	1 prospective popPK study with dosing simulations (*n* = 63) [102]	0.1–15.3 y 4.2–70 kg	Recommended age-differentiated dosing regimens lead to adequate attainment of the target AUC/MIC (>80) for sensitive pathogens Dose increase needed if MIC >1 mg/L Dose reduction needed if liver impairment (aspartate aminotransferase)
Ciprofloxacin	1	1 prospective study (*n* = 20) [103]	3 m–4.75 y 4.2–21.2 kg	No difference in CL and Vd between children aged <1 y and older. 20 mg/kg/day sufficient to cover pathogens with an MIC up to 0.8 mg/L 30 mg/kg/day in 3 doses needed in patients with normal renal function infected by pathogens with an MIC > 0.8 mg/L
Daptomycin	3	1 popPK study (*n* = 4) [104] 2 case reports (*n* = 2) [105,106]	8–14 y 17–45 kg	Higher Vd and CL in sepsis patients vs. the patient without sepsis [104] CL in sepsis patients is double the CL in non-critically ill children [104] Children with sepsis showed suboptimal AUC values, even with increased dosing. This was even more pronounced in the burn patient. Increased dosing and TDM is recommended [104].

AUC: area under the curve; CL: clearance; C_min_: trough concentration; eGFR_Schwartz_: estimated glomerular filtration rate according to the Schwartz equation; *f*T_>MIC_: time during which the free concentration exceeds the minimum inhibitory concentration; MIC: minimum inhibitory concentration; popPK: population pharmacokinetic; RCT: randomised controlled trial; TDM: therapeutic drug monitoring; Vd: distribution volume.

**Table 8 antibiotics-10-01182-t008:** Overview of all antibiotic PK studies performed in (near)term neonates supported with whole body hypothermia to treat moderate to severe perinatal asphyxia.

Reference	Antibiotic	Study Aspects	Results
Bijleveld et al. 2016 [161]	Gentamicin	Prospective popPK study during and following WBH (*n* = 47)	CL of a typical patient (3 kg, 40 weeks) was 0.06 L/kg/h and increased (+29%) after rewarming. Vd_central_ 0.46 L/kg.
Liu et al. 2009 [162]	Gentamicin	Retrospective study in asphyxia cases, either or not undergoing WBH (*n* = 55)	Impaired renal function is strongly associated with raised serum through concentrations, without additional effect of WBH versus normothermia.
Frymoyer et al. 2013a [163]	Gentamicin	Retrospective study evaluating gentamicin, 5 mg/kg q24h (*n* = 29) vs. q36h (*n* = 23)	CL 1.17 versus 1.15 L/h/70 kg. Elevated trough concentration (>2 mg/L): 38 vs. 4%.
Ting et al. 2015 [164]	Gentamicin	Retrospective study evaluating gentamicin, 2.5 mg/kg q12h, in various stages of asphyxia (*n* = 19 WBH cases vs. 15 controls)	Elimination half-life was longer in WBH cases (9.6 versus 7 h). Dose adjustments (interval extension) were more common in WBH.
Mark et al. 2013 [165]	Gentamicin	Retrospective study evaluating WBH cases (*n* = 16) compared to non-WBH cases (*n* = 7)	Elimination half-life was longer in WBH cases (9.16 versus 6.6 h), CL 0.04 vs. 0.05 L/kg/h (−25%)
Frymoyer et al. 2013b [166]	Gentamicin	Retrospective popPK study in WBH cases (*n* = 29)	Typical (3.3 kg) newborn: CL 0.034 L/kg/h, Vd 0.52 L/kg. Suggested dose 4–5 mg/kg/36 h.
Cies et al. 2018 [167]	Gentamicin	Prospective study in WBH cases (*n* = 19)	CL 2.2, SD 0.7 mL/min/kg, Vd 0.96, SD 0.4 L/kg. Suggested dose 5 mg/kg q36h.
Zahora et al. 2009 [168]	Gentamicin	Study in WBH (*n* = 12) compared to normo-thermic non asphyxia cases (*n* = 19)	CL 0.65, SD 0.23 versvs.us 0.9 (SD 0.31) mL/min/kg (−28%)
Riera et al. 2013 [169]	Gentamicin	PopPK study in WBH cases (*n* = 6), using trough and peak concentrations	Time interval extension from 24 h to 36 h needed (4 mg/kg).
Cristea et al. 2017 [170]	Amikacin	Retrospective popPK study in WBH cases (*n* = 56) and controls (*n* = 874)	CL 49.5 mL/kg/h, −40% compared to term controls, time interval from 24 h to 36 h.
Cies et al. 2017 [171]	Ampicillin	Prospective popPK study in WBH cases (*n* = 13)	CL 0.43, SD 0.12 mL/min/kg, Vd 0.52, SD 0.28 L/kg.
Bijleveld et al. 2018a [172]	Amoxicillin	Prospective popPK study in WBH cases (*n* = 125)	Typical patient (3 kg), CL increases from 0.26 to 0.41 L/h from day 2 to day 5. 50–75 mg/kg/24 h (q8h) suggested.
Bijleveld et al. 2018b [173]	Benzylpenicillin	Prospective popPK study in WBH cases (*n* = 41)	Typical patient (3 kg), CL 0.48 and Vd 0.62 L/kg.

CL: clearance; popPK: population pharmacokinetic; SD: standard deviation; Vd_(central)_: (central) distribution volume; WBH: whole body hypothermia.

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
