# Peer review of "Pharmacokinetics of Antibiotics in Pediatric Intensive Care: Fostering Variability to Attain Precision Medicine"

_antibiotics, 2021, doi:10.3390/antibiotics10101182_

Round 1

Reviewer 1 Report

Thank you for the opportunity to review this manuscript. in my opinion, this manuscript has been well written and presents a lot of interesting/important data concerning antimicrobial dosing in critically ill children. I only have several minor comments/suggestions for authors to consider: 1. The manuscript discusses PK/PD target attainment but what these targets have not been defined/discussed. For example, when discussing beta-lactams, authors need to state the PK/PD target for beta-lactams and perhaps, they can also briefly review whether these targets/exposures are similar to the adult population. In my opinion, PKPD targets should be defined for each antibiotic/antibiotic class that they have discussed in this manuscript 2. Section 4 – it will be better and structured to sub-divide this section also by antibiotic/antibiotic class. For example 4.1 Augmented renal clearance 4.1.1. Aminoglycosides 4.1.2 Beta-lactams etc. 3. PK/PD target non-attainment – did any of these studies report or link this with clinical end-point? If they did, I suggest authors to also include them in this manuscript 4. Line 321: I suggest adding higher degree of lipophilicity rather than lipophilic drugs 5. The title seems to highlight precision medicine and I was expecting more on this topic but authors only included this as discussion? I suggest to have a specific section on this and describe what are the approaches to optimise dosing via precision methods e.g. nomogram-based, linear regression-based, Bayesian, dosing software etc.

Author Response

Pharmacokinetics of antibiotics in pediatric intensive care: fostering variability to attain precision medicine (antibiotics-1381543)

Dear Editor,

Dear Reviewers,

We wish to thank the editor and the reviewers for their valuable time and for giving us the opportunity to submit a revision of our manuscript, entitled ‘Pharmacokinetics of antibiotics in pediatric intensive care: fostering variability to attain precision medicine’ in Antibiotics. We hope this revision will be considered for further review and subsequent publication.

We considered the thoughtful and constructive suggestions raised by the reviewers and addressed each one of them. The suggestions made by the reviewers changed our manuscript, for the better in our point of view. Reviewers’ comments were addressed point-by-point. Adjustments were made by rephrasing sentences, extending/adding several paragraphs,  adding an additional table (illustrating the PK/PD targets) and an abbreviations section to clarify the content of our paper and to answer the reviewers’ comments.

Please find below our detailed responses to the comments of the reviewers. We submitted simultaneously the revised manuscript (with corrections in track changes).

We hope that the editor and the reviewers will consider this revision acceptable for publication in Antibiotics and look forward to reading your comments and decision. We are glad to address any further questions or comments.

On behalf of all co-authors,

Matthias Gijsen & Karel Allegaert

Response to Reviewer 1 Comments

Thank you for the opportunity to review this manuscript. in my opinion, this manuscript has been well written and presents a lot of interesting/important data concerning antimicrobial dosing in critically ill children.

Response: Dear reviewer, we thank you for your valuable time and we appreciate your critical evaluation of our manuscript. Below, we addressed each comment point-by-point. These comments improved our manuscript.

I only have several minor comments/suggestions for authors to consider:

  1. The manuscript discusses PK/PD target attainment but what these targets have not been defined/discussed. For example, when discussing beta-lactams, authors need to state the PK/PD target for beta-lactams and perhaps, they can also briefly review whether these targets/exposures are similar to the adult population. In my opinion, PKPD targets should be defined for each antibiotic/antibiotic class that they have discussed in this manuscript

Response: We understand your comment, as we indeed mention PK/PD target attainment at several occasions. However it is difficult to assign a specific (numeric) PK/PD target to each antibiotic (class) as different studies often used somewhat different targets.

Therefore, we added an additional table (Table 2), which presents the general PK/PD index/category commonly used per antibiotic (class), i.e.,  fT>MIC, fCmax/MIC and/or AUC/MIC. These are based on review papers in adults (Abdul-Aziz et al. Int Care Med 2020; Chai et al. Pharmaceutics 2020) and pediatrics (Downes Int J Antimicrob Agents 2014). Also we specified in the text that PK/PD targets used in pediatric ICU patients are commonly the same targets as those used in adult ICU patients.

  1. Section 4 – it will be better and structured to sub-divide this section also by antibiotic/antibiotic class. For example 4.1 Augmented renal clearance 4.1.1. Aminoglycosides 4.1.2 Beta-lactams etc.

Response: Thank you, we considered your suggestion. However, considering that the text is not excessively long for this section, and the fact that the different sections will not be structured the same way then (as sub-divisions are not presented in other sections, and not always the same antibiotics are discussed in each section), we believe it is best for the uniformity of the manuscript to keep the current structure.

We might consider adding subheadings in the tables if you consider this to be of added value.

  1. PK/PD target non-attainment – did any of these studies report or link this with clinical end-point? If they did, I suggest authors to also include them in this manuscript

Response: This is indeed a relevant comment. We checked this, but until now, in children, studies focused only on the PK (i.e., the exposure). Hence, target non-attainment is considered as a surrogate marker for worse clinical outcome, while it has not been shown in large trials (ideally RCTs) that increasing target attainment (by dose optimization) leads to substantial clinical benefit.

Robust data demonstrating a clinical benefit from antibiotic dose optimization are lacking, especially in children. In adults, there are some limited data showing that decreased target attainment leads to worse clinical outcome. The most important is probably the DALI study (Roberts et al. Clin Inf dis 2014), which raised the awareness for underexposure in the ICU. Also, the EXPAT trial (Abdulla et al. Crit Care 2020) showed increased ICU length-of-stay with failure of PK/PD target attainment. However, it is still unclear if dose optimization not only leads to better exposure, but also to better clinical outcome. Currently, two large RCTs are underway (in adults), which might provide an answer to this question. For one of these, the TARGET trial, preliminary results have recently been presented at ECCMID. These preliminary results are discussed at the end of the Discussion section.

  1. Line 321: I suggest adding higher degree of lipophilicity rather than lipophilic drugs

Response: We adapted this sentence according to your suggestion.

  1. The title seems to highlight precision medicine and I was expecting more on this topic but authors only included this as discussion? I suggest to have a specific section on this and describe what are the approaches to optimise dosing via precision methods e.g. nomogram-based, linear regression-based, Bayesian, dosing software etc.

Response: We understand your comment. However, our main intention was to collect the needed information on relevant covariates to develop precision medicine concepts to the NICU and PICU population for antibiotics. This aim has been added and further stressed in the last paragraph of the introduction. The aspects mentioned by the reviewer have also been further stressed and extended in the discussion section, with an additional referral to recent reviews to this specific (methodological) topics.

Reviewer 2 Report

I wish to congratulate the authors for their extensive, detailed, and important review of the literature.

My only remark is to include the list of abbreviations at the beginning.

Author Response

Pharmacokinetics of antibiotics in pediatric intensive care: fostering variability to attain precision medicine (antibiotics-1381543)

Dear Editor,

Dear Reviewers,

We wish to thank the editor and the reviewers for their valuable time and for giving us the opportunity to submit a revision of our manuscript, entitled ‘Pharmacokinetics of antibiotics in pediatric intensive care: fostering variability to attain precision medicine’ in Antibiotics. We hope this revision will be considered for further review and subsequent publication.

We considered the thoughtful and constructive suggestions raised by the reviewers and addressed each one of them. The suggestions made by the reviewers changed our manuscript, for the better in our point of view. Reviewers’ comments were addressed point-by-point. Adjustments were made by rephrasing sentences, extending/adding several paragraphs,  adding an additional table (illustrating the PK/PD targets) and an abbreviations section to clarify the content of our paper and to answer the reviewers’ comments.

Please find below our detailed responses to the comments of the reviewers. We submitted simultaneously the revised manuscript (with corrections in track changes).

We hope that the editor and the reviewers will consider this revision acceptable for publication in Antibiotics and look forward to reading your comments and decision. We are glad to address any further questions or comments.

On behalf of all co-authors,

Matthias Gijsen & Karel Allegaert

Response to Reviewer 2 Comments

I wish to congratulate the authors for their extensive, detailed, and important review of the literature.

Response: Dear reviewer, we thank you for your valuable time and we appreciate your critical evaluation of our manuscript. Below, we addressed your remark concerning the abbreviations.

My only remark is to include the list of abbreviations at the beginning.

Response: Thank you for this suggestion. This will indeed increase the readability of our review. According to the journal’s guidelines, we included an *Abbreviations* section in the back of the manuscript (after the *Conflict of Interest* section). Please let us know if you believe this should be in the beginning of the manuscript, we can take this up with the journal office then.

We had some uncertainties on the guidelines on how to integrate this in the paper, so that we have added this list chronologically (as appeared in the publication). We remain obviously available to further adapt this if this were perceived to be appropriate.

Reviewer 3 Report

The work helped fill an important gap relating to the knowledge and the effective use of antibiotics in ICU children

The authors have done a tremendous work, their review is comprehensive and very interesting and addresses a problem of great clinical interest.

The model they proposed to improve target exposure for antibiotics in critically ill neonates and children is very interesting. I have no observations to make other than asking the authors to clarify the sentence in lines 181-184.

Author Response

Pharmacokinetics of antibiotics in pediatric intensive care: fostering variability to attain precision medicine (antibiotics-1381543)

Dear Editor,

Dear Reviewers,

We wish to thank the editor and the reviewers for their valuable time and for giving us the opportunity to submit a revision of our manuscript, entitled ‘Pharmacokinetics of antibiotics in pediatric intensive care: fostering variability to attain precision medicine’ in Antibiotics. We hope this revision will be considered for further review and subsequent publication.

We considered the thoughtful and constructive suggestions raised by the reviewers and addressed each one of them. The suggestions made by the reviewers changed our manuscript, for the better in our point of view. Reviewers’ comments were addressed point-by-point. Adjustments were made by rephrasing sentences, extending/adding several paragraphs,  adding an additional table (illustrating the PK/PD targets) and an abbreviations section to clarify the content of our paper and to answer the reviewers’ comments.

Please find below our detailed responses to the comments of the reviewers. We submitted simultaneously the revised manuscript (with corrections in track changes).

We hope that the editor and the reviewers will consider this revision acceptable for publication in Antibiotics and look forward to reading your comments and decision. We are glad to address any further questions or comments.

On behalf of all co-authors,

The work helped fill an important gap relating to the knowledge and the effective use of antibiotics in ICU children.

The authors have done a tremendous work, their review is comprehensive and very interesting and addresses a problem of great clinical interest.

Response: Dear reviewer, we thank you for your valuable time and we appreciate your critical evaluation of our manuscript. Below, we addressed your comment.

The model they proposed to improve target exposure for antibiotics in critically ill neonates and children is very interesting. I have no observations to make other than asking the authors to clarify the sentence in lines 181-184.

Response: We rephrased these sentences in the manuscript to clarify our message. We hope these sentences are now clear to the reader.
